# The Physiological Mechanism of Exogenous Melatonin on Improving Seed Germination and the Seedling Growth of Red Clover (*Trifolium pretense* L.) under Salt Stress

**DOI:** 10.3390/plants13172527

**Published:** 2024-09-08

**Authors:** Rui Liu, Ting Wang, Jiajie Wang, Di Yan, Yijia Lian, Zhengzong Lu, Yue Hong, Xue Yuan, Ye Wang, Runzhi Li

**Affiliations:** 1College of Plant Science and Technology, Beijing University of Agriculture, Beijing 102206, China; l18213779662@163.com (R.L.); wt1016826691@163.com (T.W.);; 2Beijing Key Laboratory for Agricultural Application and New Technique, Beijing 102206, China

**Keywords:** red clover, melatonin, salt stress, physiological mechanism

## Abstract

Salt stress can affect various physiological processes in plants, ultimately hindering their growth and development. Melatonin (MT) can effectively resist multiple abiotic stresses, improving plant stress resistance. To analyze the mechanism of exogenous MT to enhance salt tolerance in red clover, we conducted a comprehensive study to examine the influence of exogenous MT on various parameters, including seed germination indices, seedling morphological traits, and physiological and photosynthetic indicators, using four distinct red clover varieties (H1, H2, H3, and H4). This investigation was performed under various salt stress conditions with differing pH values, specifically utilizing NaCl, Na_2_SO_4_, NaHCO_3_, and Na_2_CO_3_ as the salt stressors. The results showed that MT solution immersion significantly improved the germination indicators of red clover seeds under salt stress. The foliar spraying of 50 μM and 25 μM MT solution significantly increased SOD activity (21–127%), POD activity, soluble sugar content, proline content (22–117%), chlorophyll content (2–66%), and the net photosynthetic rate. It reduced the MDA content (14–55%) and intercellular CO_2_ concentration of red clover seedlings under salt stress. Gray correlation analysis and the Mantel test further verified that MT is a key factor in enhancing seed germination and seedling growth of red clover under salt stress; the most significant improvement was observed for NaHCO_3_ stress. MT is demonstrated to improve the salt tolerance of red clover through a variety of mechanisms, including an increase in antioxidant enzyme activity, osmoregulation ability, and cell membrane stability. Additionally, it improves photosynthetic efficiency and plant architecture, promoting energy production, growth, and optimal resource allocation. These mechanisms function synergistically, enabling red clover to sustain normal growth and development under salt stress.

## 1. Introduction

Red clover (*Trifolium pretense* L.), an environmentally and economically significant green manure crop [1], has great potential for cultivation and promotion throughout China, especially in the northwest and southwest regions. It has a strong nitrogen-fixing capacity, which can reduce the reliance on chemical nitrogen fertilizers in agricultural production. At the same time, as a kind of green manure, it can be turned and buried into the soil to increase soil fertility, and its root system can penetrate deep into the soil, which can help to loosen the soil and increase the permeability, thus improving the soil structure [2,3]. In recent years, China has placed greater emphasis on researching red clover germplasm resources to develop new varieties of red clover that offer enhanced ecological and economic benefits [4,5]. Given the natural susceptibility of red clover to salt stress, a thorough investigation into its salt resistance is crucial [6].

In recent years, the global expansion of saline-alkali land areas and the escalating severity of soil salinization have become pressing issues [7,8]. Saline-alkali soil in China is mainly found in the northwest [9], inland territories of northern China, and coastal beaches [10]. Particularly in the north, the climatic characteristics of low rainfall and high evaporation result in a tendency for soluble salts in the soil to accumulate in the surface layer, leading to soil salinization [11,12,13]. These salts include Cl^−^, SO_4_^2−^, CO_3_^2−^, HCO_3_^−^, Na^+^, K^+^, Ca^2+^ and Mg^2+^. Common neutral salts found in salt stress are NaCl and Na_2_SO_4_, while alkali salts include NaHCO_3_ and Na_2_CO_3_. Understanding the damage caused to plants by different pH types of salt stress can provide a theoretical basis for improving salt tolerance.

Salt stress causes damage to the structure and function of plant cell membranes, increased membrane permeability, and the extravasation of intracellular substances (e.g., electrolytes, organic matter, etc.), thus affecting the normal function of cells. It also induces the excessive production of reactive oxygen species (ROS) in plants, leading to membrane lipid peroxidation and further damage to cell membranes [14,15,16]. Red clover responds with enzyme system, including SOD and POD for ROS scavenging [17,18]. Proline, soluble sugar, and MDA content indicate cytoplasmic homeostasis. CO_2_ concentration, photosynthetic rate, stomatal conductance, transpiration rate, and chlorophyll content are used to assess salt tolerance [19,20]. Developing saline–alkali-tolerant red clover varieties is crucial but time-consuming [21]. Exogenous plant growth regulators mitigate salt and alkali effects, enhance plant resistance, and maintain plant growth [22].

Melatonin (MT), chemically referred to as N-acetyl-5-methoxytryptamin, is an indole hormone commonly associated with the pineal gland [23,24]. Its widespread presence in plants plays a crucial role in regulating responses to both biotic and abiotic stresses [25,26]. Research has demonstrated that MT possesses resistance to various abiotic stresses, including high temperatures, salinity, and heavy metals, which can negatively affect plant growth and development [27,28]. MT can reduce the accumulation of ROS, increase the activity of antioxidant enzymes, and increase osmoregulatory substances, thus protecting plants from stress-induced damage, as well as lowering chlorophyll degradation and delaying leaf senescence [29,30,31,32]. As a result, significant research efforts have focused on exploring the potential benefits of externally applying MT to mitigate the inhibitory effects of abiotic stressors on plant growth and development [33,34,35,36].

There are many studies on exogenous MT to alleviate the effects of abiotic stresses on plant growth and development, but there are limited studies on the effects of exogenous MT on red clover, and previous studies mainly focused on a single material and a single stress condition. Therefore, in this study, the responses of different varieties of red clover to different types of salt stress were compared, including not only NaCl (the most used) but also NaHCO_3_, Na_2_CO_3_, and Na_2_SO_4_, and the optimal concentration of melatonin was determined, thus revealing the potential mechanism of melatonin to improve the salt tolerance of red clovers and providing practical guidance for the salt stress management of red clovers. The results of this study not only provide new insights for improving the salt tolerance of red clover but also provide a theoretical basis for the use of plant growth regulators to alleviate salt stress.

## 2. Results

### 2.1. Effects of Exogenous MT on Seed Germination of Red Clover under Salt Stress

To assess the effects of MT on red clover seed germination under salt stress, we measured the germination rate (GR), germination potential (GP), germination index (GI), mean germination days (MGT), root length (RL), and vigor index (VI) of red clover seeds under different treatments (Table 1). Compared with the water control (CK), red clover seed germination was inhibited, and the growth cycle was prolonged under salt stress. Compared with the salt-treated groups (S), 100 μM MT imbibition significantly improved the germination of red clover seeds under salt stress and to different degrees for different varieties.

Compared with salt stress, the MT treatment significantly increased GR by 11–36%, GP by 3–17%, GI by 24–176%, RL by 17–100%, and VI by 63–284%. Under NaCl stress, the GR and VI of HI significantly increased by 28% and 172%, respectively, and the GR of H4 reached 80% after MT treatment. Under Na_2_SO_4_ stress, the GR and GI of H4 significantly increased by 29% and 124%, respectively, and the RL doubled after MT treatment. Under NaHCO_3_ stress, the GI and VI of H2 significantly increased by 108% and 222%, respectively, after MT treatment. Under Na_2_CO_3_ stress, the GR, GI, and VI of H2 were significantly increased by 36%, 176%, and 284%, respectively, after MT treatment. The results showed that MT soaking alleviated the inhibitory effect of salt stress on red clover seed germination to a certain extent and promoted root growth.

### 2.2. Effects of Exogenous MT on Morphological Characteristics of Red Clover Seedlings under Salt Stress

To further analyze the effect of MT on the growth of red clover seedlings under salt stress, the morphological characteristics of red clover seedlings were determined under different treatments (Figure 1 and Appendix A). Compared with CK, the growth of red clover seedlings was inhibited, and the plants were short under salt stress. However, the spraying of MT improved the plant shape with a significant increase in plant height, leaf length, and leaf width, which alleviated the inhibition of salt stress, and the plant height of red clover seedlings significantly increased by 8–18%. Under NaCl and Na_2_SO_4_ stress, the plant height of H1 was significantly reduced by 39% and 31%, respectively, and significantly increased by 18% and 14% after spraying 50 μM MT, which delayed the inhibition of plant growth. Under NaHCO_3_ stress, the plant height of H3 was significantly improved by 15% after spraying 25 μM MT. Under Na_2_CO_3_ stress, H2 and H3 plant heights were significantly reduced by 39% and 32%, respectively, and MT treatment significantly increased H2 by 14% and H3 by only 8% (Figure 1). The results showed that the foliar spraying of melatonin delayed the growth inhibition of red clover seedlings by salt stress.

### 2.3. Effects of Exogenous MT on Osmoregulation and the Cell Membrane Structure of Red Clover Seedlings under Salt Stress

To further investigate the effects of exogenous MT on osmoregulation in red clover seedlings under salt stress, changes in osmoregulatory substances in red clover seedlings were determined under different treatments. Compared with CK, soluble sugar and proline contents were significantly increased under salt stress, and the accumulation of these substances helped the plants to maintain osmotic homeostasis; the increase in MDA and relative conductivity reflected the increase in cell membrane permeability, which is usually due to the disruption of the cell membrane structure and the damage of the plant cell membrane, resulting in membrane lipid peroxidation. Spraying MT significantly increased the soluble sugar and proline contents, and the MDA and relative conductivity contents significantly decreased, which improved the cellular osmoregulatory capacity and alleviated the osmotic stress induced by salt stress.

Compared with salt stress, melatonin treatment significantly increased the soluble sugar and proline contents of red clover seedlings by 6–51% and 22–117%, respectively, and the MDA contents were significantly reduced by 14–55%. Under NaCl and Na_2_SO_4_ stress, the soluble sugar and proline contents of H1 significantly increased by 20% and 18%, and 92% and 111%, respectively, and the MDA contents of H4 were significantly reduced by 55% and 35% after spraying 50 μM MT. Under NaHCO_3_ and Na_2_CO_3_ stress, the proline contents of H3 were significantly increased by 117% and 114% after spraying 25 μM MT, respectively (Figure 2 and Appendix A). The results showed that the foliar spraying of melatonin could increase the accumulation of osmoregulatory substances to maintain osmotic balance in red clover seedlings under salt stress, thus delaying the inhibition of plant growth by salt stress.

### 2.4. Effects of Exogenous MT on Antioxidant Enzyme Activities of Red Clover Seedlings under Salt Stress

To further investigate the effect of exogenous MT on the antioxidant regulation of red clover seedlings under salt stress, the activities of the antioxidant enzymes SOD and POD in red clover seedlings were determined under different treatments (Figure 3). Compared with CK, the antioxidant enzyme activities of red clover seedlings were significantly increased under salt stress, which is an important way for plants to adapt to salt-stressed environments, helping them to resist stress and reduce the damage caused by salt stress. The SOD and POD activities of red clover seedlings were further increased by MT treatment under salt stress, which enhanced the antioxidant capacity, scavenged ROS, and reduced oxidative damage.

Under NaCl stress, the SOD and POD activities of H1 were significantly increased by 64% and 38%, respectively, after the spraying of 50 μM MT, which reduced the damage of ROS to the cell membranes, maintained the normal cellular function, and reduced the damage to seedling growth. Under Na_2_SO_4_ stress, MT treatment significantly increased the SOD and POD activities of H4 by 127% and 46%, respectively. Under NaHCO_3_ stress, the SOD and POD activities of H2 were significantly increased by 174% and 11%, respectively, after spraying 25 μM MT. Under Na_2_CO_3_ stress, the SOD and POD activities of H1 were significantly increased by 136% and 22%, respectively, after MT treatment. The results showed that the foliar spraying of melatonin could effectively regulate the antioxidant capacity of red clover seedlings under salt stress, reduce oxidative damage, and improve salt tolerance.

### 2.5. Effects of Exogenous MT on Photosynthetic Parameters of Red Clover Seedlings under Salt Stress

To investigate the effect of MT on photosynthesis in red clover seedlings under salt stress, the photosynthetic parameters of red clover seedlings were determined under different treatments. Compared with CK, the intercellular CO_2_ concentration significantly increased, whereas other photosynthetic parameters significantly decreased under salt stress, leading to a decrease in the photosynthetic rate and a slowdown in plant growth. However, MT spraying significantly increased the chlorophyll content, the net photosynthetic rate, stomatal conductance, and the transpiration rate and reduced the intercellular CO_2_ concentration of red clover seedlings under salt stress, thus improving photosynthetic efficiency (Figure 4 and Appendix A).

Compared with salt stress, melatonin treatment significantly increased the chlorophyll contents by 2–66% and significantly reduced intercellular CO_2_ concentration by 15–42%. Under NaCl and Na_2_SO_4_ stress, the chlorophyll content of H2 was significantly increased by 65% and 25%, and the intercellular CO_2_ concentration was reduced by 27% and 24% after spraying 50 μM MT, respectively. Under NaHCO_3_ stress, the chlorophyll content of H1 was significantly increased by 36%, and the intercellular CO_2_ concentration was significantly reduced by 40% after 25 μM MT treatment. Under Na_2_CO_3_ stress, the chlorophyll content of H3 was significantly increased by 23%, and the intercellular CO_2_ concentration was significantly reduced by 42% after MT treatment (Figure 4). The results showed that the foliar spraying of melatonin could regulate the stomatal movement and increase the chlorophyll content of red clover seedlings under salt stress, thus increasing the photosynthetic rate and reducing the inhibition of seedling growth.

### 2.6. Comprehensive Analysis of the Effects of Exogenous MT on Red Clover Seed Germination and Seedling Growth

To comprehensively evaluate the regulatory effect of MT on red clover seed germination and seedling growth under salt stress, a gray correlation analysis was performed on the seed germination indicators and the physiological parameters of red clover seedlings after MT treatment under four salt stresses (Figure 5A). The results showed that MT promoted the seed germination and seedling stages of red clover under all four salt stresses, but the degree of improvement was different for different salt stresses; the gray correlation coefficient was the largest under NaHCO_3_ stress, indicating that the delayed effect of MT on this stress was more obvious.

To further analyze the factors affecting salt tolerance in red clover, the Mantel test was conducted to characterize the intricate networks of the correlations among two influencing factors (MT and variety) and twenty indicators of red clover (Figure 5B). The results showed that MT was a key factor in improving seed germination and the seedling growth of red clovers under salt stress, and most of the indicators were significantly and positively correlated with MT treatment. At the same time, the correlation with varietal differences was lower. The results further verified that 100 μM MT immersion significantly enhanced red clover seed germination under different salt stresses; foliar spraying with 50 μM MT significantly enhanced red clover seedling growth under NaCl and Na_2_SO_4_ stresses; and 25 μM MT significantly enhanced red clover seedling growth under NaHCO_3_ and Na_2_CO_3_ stresses. MT can effectively improve the salt tolerance of red clover under different salt stresses to maintain normal growth and development in a salt-stressed environment.

## 3. Discussion

### 3.1. Exogenous MT Improved the Antioxidant and Osmoregulatory Capacity of Red Clover under Salt Stress

Salt stress can have some negative effects on plant growth and development, including seed germination, seedling growth, root development, and photosynthesis [37,38,39,40]. High concentrations of salt lead to an increase in osmotic pressure, making it difficult for seeds to absorb water [41]; excessive Na^+^ and Cl^−^ enter the seeds, interfering with intracellular ionic balance and damaging cellular structure and function [42]; and the excessive intracellular production of reactive oxygen species (ROS) leads to lipid peroxidation, affecting the permeability and selectivity of the cell membrane, which then affects the absorption of water and nutrients in the seeds and inhibits seed germination and seedling growth [43,44] Melatonin plays an important role in mitigating the negative effects of abiotic stress on plants. The application of appropriate concentrations of MT can effectively improve seed germination under stress [45,46], increase biomass after germination, reduce damage from salt stress, promote plant growth, and enhance resilience [47,48].

Previous studies have elucidated that melatonin improves plant growth through multiple pathways under abiotic stress [49,50]. Among them, melatonin enhances antioxidant enzyme activity, which scavenges ROS, reduces lipid peroxidation, protects the integrity of cell membranes [51,52], and delays the effects of stress on plant growth and development [53]. ROS are produced by abiotic stresses (e.g., pesticides, salinity, and drought) and biotic stresses (e.g., phytopathogens), which can lead to cell damage and death. Antioxidant enzymes have a universal role in mitigating different abiotic and biotic stresses and can scavenge ROS, thereby protecting cells from damage [54,55]. It was found that both humic acid and nitrophenol effectively increased CAT activity in wheat under fungal infestation and alleviated oxidative stress associated with biotic stresses [54]. Wang et al. found that the activities of antioxidant enzymes (CAT and SOD) in wheat seedlings increased after MT spraying, scavenging reactive oxygen species and thus alleviating the inhibition of seed germination and seedling growth by salt stress [17]. Wang et al. found that 100 μM MT soaking enhanced the activities of antioxidant enzymes and non-enzymatic antioxidants in oat seeds, affected the AsA-GSH cycle, enhanced oat salinity tolerance, and delayed the inhibition effect [51]. Our results were consistent with the above. Different types of salt stress caused an increase in reactive oxygen species (ROS), exacerbated membrane lipid peroxidation, and inhibited the growth of red clover seedlings, in which the activity of SOD fluctuated greatly, and SOD could disproportionate the free radicals O^2−^ to H_2_O_2_ and O_2_, thus reducing the accumulation of free radicals and mitigating the degree of membrane lipid peroxidation. MT improves the resistance of red clover seedlings by increasing the activity of the antioxidant enzyme system in red clover seedlings, thereby scavenging reactive oxygen molecules, protecting the cell membranes from oxidative damage, and maintaining normal plant growth and development. However, the response of MT to different varieties of materials and different types of salt stress is also different, so it is necessary to choose the appropriate MT concentration according to the materials and stresses.

Salt stress causes an increase in the osmotic pressure of the soil solution, making it difficult for plants to absorb water, resulting in osmotic stress and thus reducing plant resistance [56,57,58]. Previous studies have shown that melatonin promotes the accumulation of osmoregulatory substances in plants, maintains intracellular osmotic homeostasis, and attenuates the water loss caused by external high-salt environments [59,60,61]. The production of osmoregulatory substances is an important way for plants to avoid abiotic stress [62,63] and MT plays a role in osmoregulation against salt-induced damage by increasing the accumulation of substances such as proline and soluble sugars [49,64,65]. In our study, MDA and relative conductivity responded better to the degree of damage to the plasma membrane structure and oxidative homeostasis, and the exchange of substances between inside and outside the cell increased under salt stress, leading to an increase in relative conductivity. However, MT treatment significantly increased soluble sugar and proline content, decreased osmotic potential, and maintained cellular osmotic balance. The MDA content and relative conductivity decreased to different degrees, which helped to protect the integrity of the cell membrane and maintain the stability of the intracellular environment, thus improving the stress tolerance and growth ability of red clover seedlings (Figure 2 and Appendix A).

### 3.2. Exogenous MT Attenuates the Inhibition of Photosynthesis in Red Clover Seedlings by Salt Stress

During plant growth, leaves change to adapt to challenging environments [66,67,68,69]. Leaf stomata play a key role in water and gas exchange and have a major impact on transpiration [70,71,72] respiration, and photosynthesis in plants [73,74]. Studies have shown that salt stress leads to the closure of stomata in plant leaves, limiting the entry of CO_2_ [75,76], weakening transpiration, decreasing the chlorophyll content, and resulting in the yellowing of leaves, which in turn reduces the absorption and utilization of light energy by the plant, decreases photosynthesis, and hinders plant growth [77,78,79].

Previous studies found that melatonin-treated plants had higher chlorophyll content; increased photosynthetic pigment content and photosynthetic rate; and increased activity of photosynthetic enzymes such as Rubisco and ATP synthase, which protected the photosynthetic system from salt-induced damage and maintained growth [80]. Zhang et al. found that melatonin increased the photosynthetic rate of plants under drought stress, and this improvement was attributed to the upregulation of key photosynthetic enzymes such as Rubisco and the maintenance of chlorophyll content [81]. Melatonin also plays a role in reducing oxidative damage to the photosynthetic system, as evidenced by reduced MDA levels and increased antioxidant enzyme activity, further contributing to the protection of photosynthetic function [82]. This effect was also verified in our study, where MT-treated plants showed a decrease in MDA and relative conductivity, an increase in SOD and POD activities, an increase in antioxidant capacity, and a decrease in the degree of damage to the photosynthetic system (Figure 2, Figure 3 and Figure 4). Under salt stress, stomatal closure inhibited further gas exchange, and the increased intercellular CO_2_ concentration led to a decrease in the rate of photosynthesis, and therefore the chlorophyll content decreased, leading to the yellowing of the leaves and a reduction in the plant’s ability to absorb light energy, thus diminishing the efficiency of photosynthesis. The MT-treated plants showed significant improvement in photosynthetic indexes, and photosynthesis and transpiration increased (Figure 4 and Appendix A), which were interdependent and regulated the water balance of the plant, promoted the absorption and transport of water and nutrients, alleviated the effects of stress on plant metabolism, and maintained the normal growth and development of the plant.

### 3.3. Limitations and Perspectives

It was also found that melatonin regulates the levels of phytohormones to alleviate salt stress-induced seed germination and plant growth [83,84,85]. MT promotes seedling growth and alleviates stress-induced inhibition by regulating the balance of phytohormones, for instance, by decreasing abscisic acid (ABA) levels and increasing gibberellin (GA) levels [86]. MT also upregulates stress-responsive genes, including the genes encoding antioxidant enzymes and osmoregulation, thereby enhancing the stress tolerance of seedlings [87]. Although our study provides valuable insights into the mechanisms by which melatonin promotes seedling growth under salt stress, there are some limitations. For example, phytohormones are essential signaling molecules for plants to sense changes in the external environment, regulate their growth status, resist adverse environments, and maintain survival, and they are of great importance for regulating various growth and development processes and environmental responses in plants [88,89]. The mechanism by which melatonin regulates the growth of red clover seedlings has not yet involved changes in phytohormone levels under salt stress, and the exact molecular signaling pathways involved in mediating stress relief remain to be fully elucidated. Therefore, future studies need to further understand the characteristics of hormone level changes and identify the specific signaling pathways and genes involved in MT-mediated stress relief to gain insight into the molecular basis of these effects. In addition, the multifaceted protective effects of MT on seedling growth conditions highlight its potential as a growth regulator under stress, but the optimal concentration and specific application methods vary depending on plant species and stress severity, and subsequent field studies are needed to validate its effects.

## 4. Materials and Methods

### 4.1. Test Materials and Reagents

Four red clover seed materials, designated as H1, H2, H3, and H4, from the provinces of Hubei, Jiangsu, Sichuan, and Gansu, were used in this study. These seeds were harvested in 2021 and saved for two years. Neutral salts, including NaCl and Na_2_SO_4_, as well as alkaline salts NaHCO_3_ and Na_2_CO_3_, were employed for the salt stresses. MT was procured by Bio Basic Inc. (BBI, Shanghai, China), while other chemicals were acquired from Sinopharm Chemical Reagent Beijing Co., Ltd. (Beijing, China). All chemical reagents used in the experiments were found to meet analytical-grade standards.

### 4.2. Experimental Design

#### 4.2.1. Seed Germination Test

The seeds were soaked in MT (100 μM) for 4 h after disinfection. Two sheets of filter paper were lined in the germination boxes, and 50 seeds were planted in each box, with each treatment being replicated three times. The prepared solutions of NaCl (150 mM), Na_2_SO_4_ (100 mM), NaHCO_3_ (75 mM), and Na_2_CO_3_ (30 mM) (8 mL each) were used to saturate the filter papers, after which they were cultured in an incubator set to 25 °C with a 16 h light cycle.

#### 4.2.2. Seedling Growth Test

After 14 days of seedling growth, distilled water was replaced by a solution containing NaCl (150 mM), Na_2_SO_4_ (100 mM), NaHCO_3_ (75 mM), and Na_2_CO_3_ (30 mM). This salt solution was applied every 4 days, and the salt pouring was stopped after 14 days. Seedlings were sprayed with 50 μM MT treatment under NaCl and Na_2_SO_4_ stress, whereas the seedlings were sprayed with 25 μM MT treatment under NaHCO_3_ and Na_2_CO_3_ stress. Morphological, physiological, biochemical, and photosynthetic parameters were measured after seven days of continuous MT spraying, and each treatment was repeated three times.

### 4.3. Measurement Items and Methods

#### 4.3.1. Determination of Seed Germination Indicators

Seed germination was measured daily, and germination was defined as the exposure of the radicle to half of the seed coat. The germination rate (GR) and germination potential (GP) were assessed on days 2 and 7 after initiation, respectively. The GP was calculated as GP (%) = (number of seeds germinated on day 2/total number of experimental seeds) *×* 100. The GR was calculated as GR (%) = (number of seeds germinated on day 7/total number of experimental seeds) *×* 100. The germination index (GI) was calculated as GI = Σ(Gt/Tt), where Gt is the number of germinated seeds per day corresponding to Tt, and Tt is the day of the germination test. The vigor index (VI) was calculated as VI = GI *×* S, where S is the root length of germinated seeds on day 7.

#### 4.3.2. Determination of Physiological and Biochemical Indexes of Seedlings

The activity of superoxide dismutase (SOD) was assessed using the nitroblue tetrazolium method [90] Leaves weighing 0.2 g were taken and ground into pulp; then, 2 mL of phosphate buffer was added, and the mixture was centrifuged at 4 °C, 10,000 r/min for 10 min to obtain the crude SOD enzyme solution. From the crude SOD enzyme solution, 100 μL was taken and mixed with 3.9 mL of the reaction mixture (1 mg/mL EDTA-Na_2_, 0.1 mg/mL riboflavin, 20 mg/mL L-methionine, and 1 mg/mL NBT), and the mixture was placed in a 25 °C, 4000 Lx light incubator to react for 30 min. After the reaction, the light was avoided, and the absorbance was measured at 560 nm.

Peroxidase (POD) activity was measured using the guaiacol method [91]. Briefly, 40 μL of crude enzyme extract was taken and added to 3 mL of the reaction solution (0.2 mol/L phosphate buffer, guaiacol, 30% H_2_O_2_), and the absorbance value was measured at 470 nm over 1 min.

The contents of malondialdehyde (MDA) and soluble sugar were determined using the thiobarbituric acid colorimetric method [92]. Leaves weighing 0.4 g were taken and ground into pulp; TCA was added, and the mixture was centrifuged at 4 °C and 4000 r/min for 10 min. Then, 2 mL of the supernatant was taken, 2 mL of 0.6% TBA solution was added, and the mixture was well mixed and boiled for 15 min. After cooling rapidly, the mixture was centrifuged again. An appropriate amount of the supernatant was taken, and the upper absorbance values were determined at 450 nm, 532 nm, and 600 nm.

The proline content was determined with the ninhydrin method [93]. Leaves weighing 0.8 g were taken and ground into pulp; then, 8 mL of 3% sulfosalicylic acid solution was added, and the mixture was boiled and extracted for 10 min and then cooled and filtered to obtain the extract. Afterward, 2 mL of the extract was taken, 2 mL of glacial acetic acid and 2 mL of acid ninhydrin reagent were added, and the mixture was boiled in a boiling water bath for an additional 30 min, resulting in a red solution. After cooling, 4 mL of toluene was added and mixed, followed by static stratification. The upper layer of the solution was subjected to centrifugation at 3000 r/min for 5 min, and the absorbance value was determined at 520 nm using the upper layer of the centrifuged solution.

The relative conductivity was determined using the immersion method [94]. Fresh leaves were taken, cut into pieces, weighed at 0.1 g, and put into a 10 mL clean centrifuge tube. The tube was left for 20 h, during which time it was frequently shaken up and down, and the conductivity was measured as R1 with a conductivity meter. After boiling the sample in a boiling water bath for 30 min, the conductivity was measured as R2 after cooling, and the relative conductivity was calculated.

The chlorophyll content was determined using a BERC-502 handheld chlorophyll meter, and the photosynthetic indexes were determined with the CIRAS-3 portable photosynthesis/fluorescence analysis system.

### 4.4. Statistical Analysis

The experimental data were classified and computed using Microsoft Excel 2013. The data were statistically analyzed using IBM SPSS Statistics SPSS 23.0, subjected to gray correlation analysis and the Mantel test correlation heatmap using RStudio, and model plotting was performed using Figdraw.

## 5. Conclusions

In this study, the effects of exogenous melatonin on seed germination and the seedling growth of red clover were investigated by simulating a salt-stress environment. The potential mechanism of melatonin to improve salt tolerance in red clover was revealed (Figure 6), considering the following factors: (1) improving osmoregulation ability and cell membrane permeability and reducing the degree of cell membrane peroxidation; (2) improving antioxidant capacity, reducing the accumulation of ROS, and mitigating the degree of oxidative damage; (3) increasing the leaf chlorophyll content, the net photosynthetic rate, stomatal conductance, and the transpiration rate, and decreasing the concentration of intercellular CO_2_ concentration, thus improving photosynthetic efficiency; and (4) improving the plant structure and reducing the inhibition of plant growth by salt stress. These mechanisms work together to enhance the salt tolerance of red clover, enabling it to maintain normal growth and development in salt-stressed environments. The results of this study not only provide new insights for an improvement in salt tolerance in red clover but also provide a theoretical basis for the use of plant growth regulators to alleviate saline–alkali stress. In addition, we also determined the optimal concentration of melatonin, which provides practical guidance for salt stress management in red clover. Future research can further explore the mechanism of melatonin in different plant species and environmental conditions, as well as its application potential in agricultural production.

## Figures and Tables

**Figure 1 plants-13-02527-f001:**
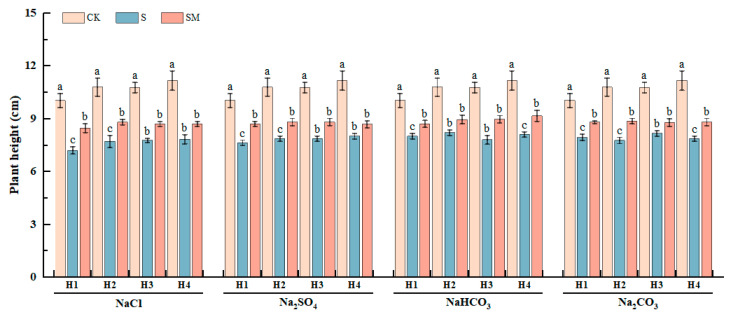
Effects of exogenous MT on plant height of red clover seedlings under different salt stresses. The error bar in the figure represents the standard deviation (SD, n = 3). Different letters indicate significant differences by Tukey’s test (*p* < 0.05).

**Figure 2 plants-13-02527-f002:**
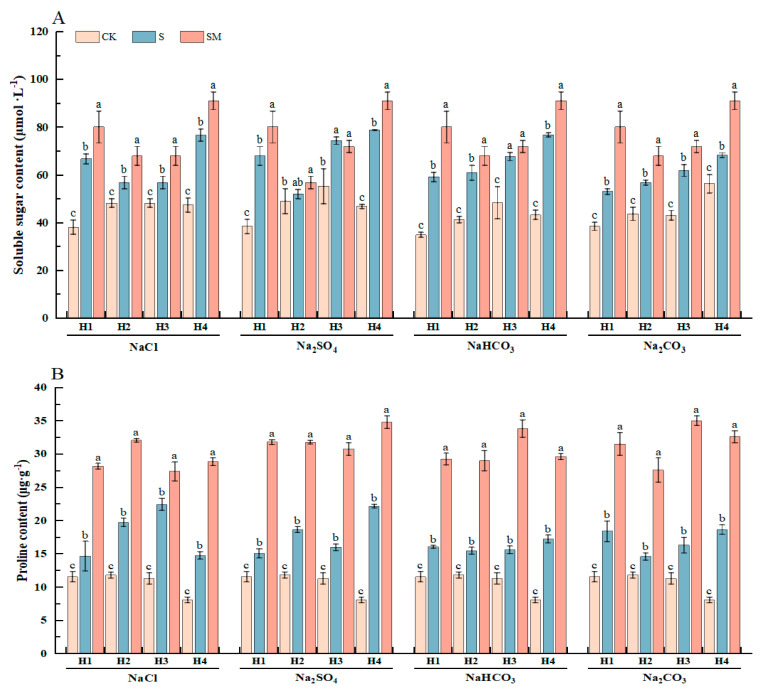
The effect of exogenous MT on osmotic substances of red clover seedlings under different types of salt stress: (**A**) changes in soluble sugar content under different treatments; (**B**) changes in proline content under different treatments; (**C**) changes in MDA content under different treatments. The error bar in the figure represents the standard deviation (SD, n = 3). Different letters indicate significant differences by Tukey’s test (*p* < 0.05).

**Figure 3 plants-13-02527-f003:**
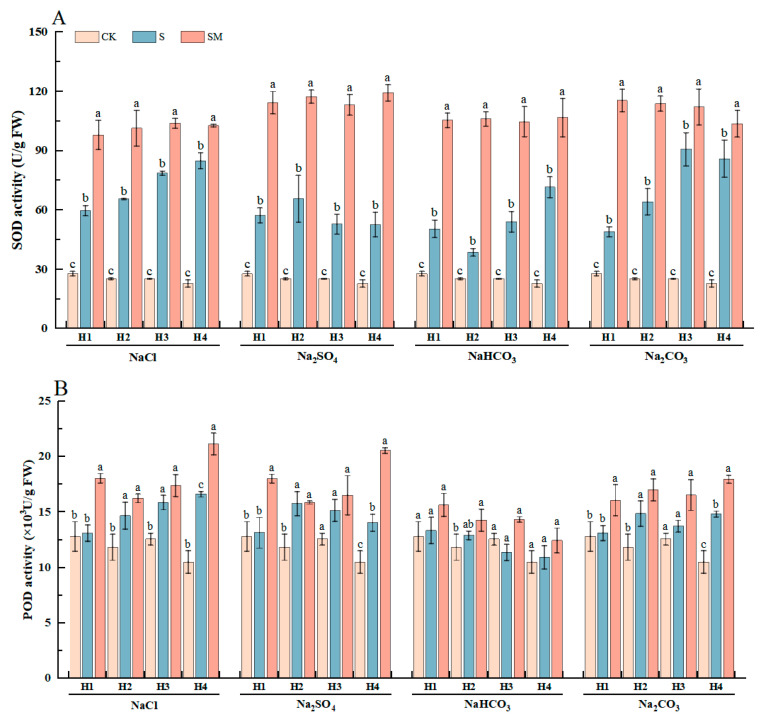
The effect of exogenous MT on antioxidant enzyme activity of red clover seedlings under different types of salt stresses: (**A**) changes in SOD activity under different treatments; (**B**) changes in POD activity under different treatments. The error bar in the figure represents the standard deviation (SD, n = 3). Different letters indicate significant differences by Tukey’s test (*p* < 0.05).

**Figure 4 plants-13-02527-f004:**
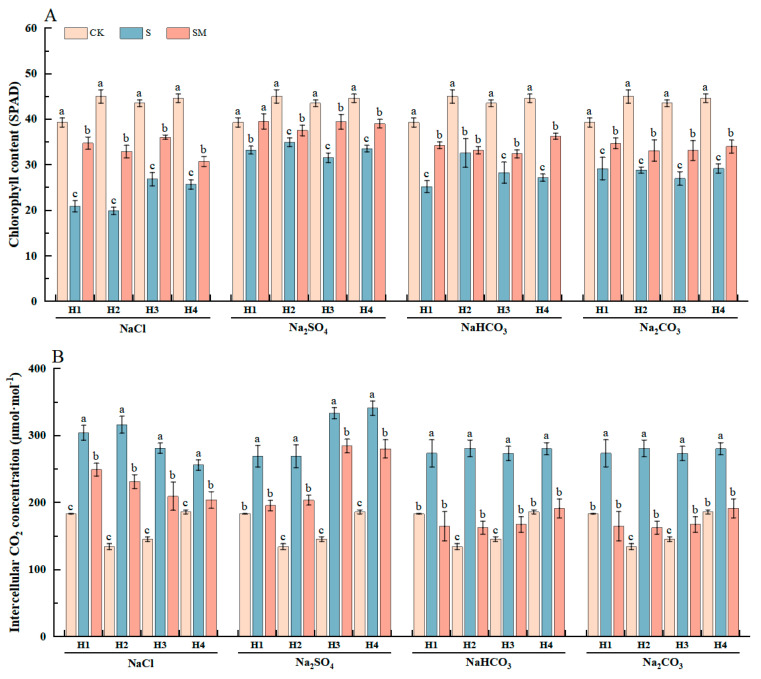
The effect of exogenous MT on photosynthetic characteristics of red clover seedlings under different types of salt stress: (**A**) changes in chlorophyll content under different treatments; (**B**) changes in intercellular CO_2_ concentration under different treatments. The error bar in the figure represents the standard deviation (SD, n = 3). Different letters indicate significant differences by Tukey’s test (*p* < 0.05).

**Figure 5 plants-13-02527-f005:**
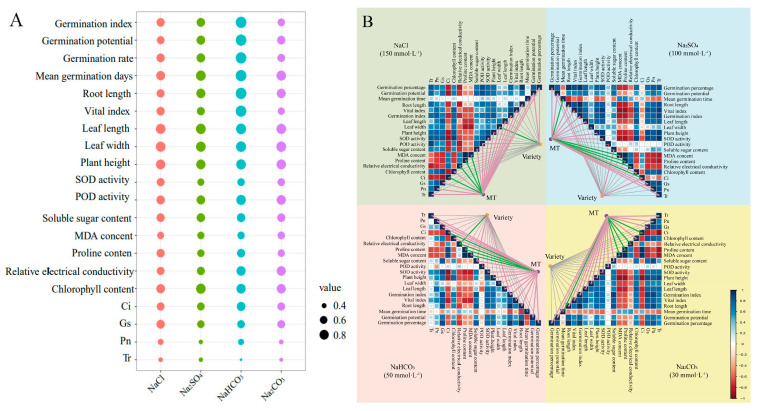
A comprehensive evaluation of MT effects on red clover seedlings under salt stress: (**A**) gray correlation coefficients of 20 indicators under four types of salt stress in red clover; (**B**) Mantel test correlation heatmap for intragroup and intergroup correlation analyses. Purple lines indicate a significant positive correlation, and green lines indicate a significant negative correlation.

**Figure 6 plants-13-02527-f006:**
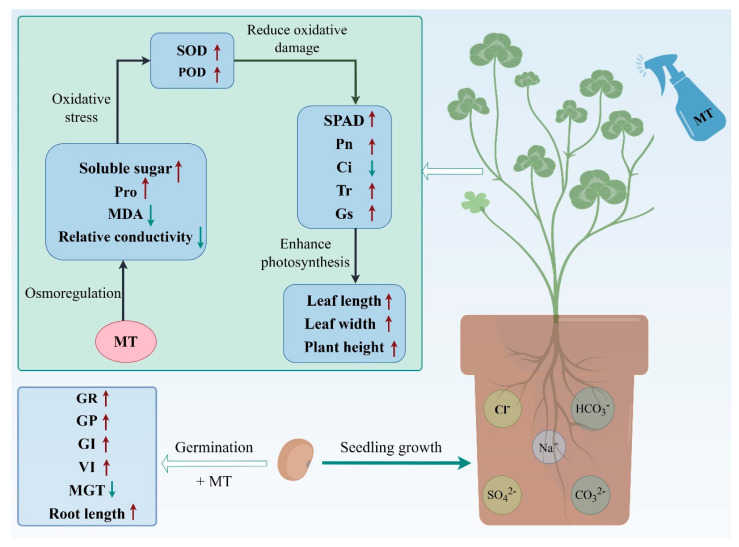
The response mechanism of exogenous MT in regulating seed germination and seedling growth in red clover under salt stress. Red arrows indicate increases, and green arrows indicate decreases after MT treatment.

**Table 1 plants-13-02527-t001:** Effect of MT on germination indicators of red clover seeds under different salt stresses.

Type	Variety	Treatment	GR/%	GP/%	MGT/d	GI	RL/cm	VI
NaCl (150 mM)	H1	CK	93 ± 3 a	62 ± 2 a	1.4 ± 0.1 b	38.3 ± 0.9 a	5.4 ± 0.1 a	207.8 ± 3.2 a
		S	34 ± 3 c	5 ± 1 c	1.6 ± 0.1 b	13.3 ± 0.8 c	1.4 ± 0.1 c	19.0 ± 0.6 c
		SM	62 ± 2 b	22 ± 3 b	2.1 ± 0.2 a	19.3 ± 1.5 b	2.7 ± 0.2 b	51.6 ± 7.4 b
	H2	CK	92 ± 2 a	62 ± 2 a	1.5 ± 0.1 b	37.4 ± 0.5 a	5.4 ± 0.3 a	201.4 ± 14.6 a
		S	47 ± 2 c	11 ± 1 b	2.0 ± 0.2 a	15.8 ± 0.8 c	1.2 ± 0.0 c	19.2 ± 1.4 c
		SM	62 ± 2 b	17 ± 3 b	2.0 ± 0.1 a	21.2 ± 0.4 b	2.1 ± 0.1 b	43.9 ± 0.6 b
	H3	CK	88 ± 2 a	61 ± 4 a	1.4 ± 0.0 a	36.8 ± 1.4 a	5.2 ± 0.2 a	192.0 ± 12.6 a
		S	40 ± 6 c	18 ± 2 c	2.0 ± 0.3 a	12.6 ± 1.4 c	1.3 ± 0.0 c	16.2 ± 2.5 c
		SM	61 ± 4 b	32 ± 2 b	2.0 ± 0.3 a	18.8 ± 1.2 b	2.0 ± 0.1 b	36.6 ± 2.0 b
	H4	CK	93 ± 2 a	68 ± 2 a	1.3 ± 0.0 b	39.8 ± 0.3 a	5.1 ± 0.2 a	202.2 ± 5.0 a
		S	64 ± 6 b	24 ± 3 c	1.9 ± 0.2 a	21.5 ± 2.7 c	1.4 ± 0.1 c	29.5 ± 5.3 c
		SM	80 ± 4 a	36 ± 2 b	1.9 ± 0.1 a	26.6 ± 1.1 b	1.8 ± 0.1 b	48.3 ± 2.4 b
Na_2_SO_4_ (100 mM)	H1	CK	93 ± 3 a	62 ± 2 a	1.4 ± 0.1 b	38.3 ± 0.9 a	5.4 ± 0.1 a	207.8 ± 3.2 a
		S	31 ± 1 c	10 ± 2 c	2.2 ± 0.1 a	9.3 ± 0.5 b	0.7 ± 0.0 c	6.2 ± 0.7 c
		SM	46 ± 7 b	18 ± 2 b	1.9 ± 0.2 a	15.4 ± 2.7 b	0.9 ± 0.1 b	14.1 ± 2.5 b
	H2	CK	92 ± 2 a	62 ± 2 a	1.5 ± 0.1 b	37.4 ± 0.5 a	5.4 ± 0.3 a	201.4 ± 14.6 a
		S	30 ± 2 c	6 ± 3 c	2.3 ± 0.2 a	8.8 ± 1.4 c	0.6 ± 0.0 b	5.4 ± 0.8 b
		SM	47 ± 2 b	15 ± 3 b	1.8 ± 0.2 ab	16.2 ± 2.1 b	1.1 ± 0.0 b	17.0 ± 1.6 b
	H3	CK	88 ± 2 a	61 ± 4 a	1.4 ± 0.0 b	36.8 ± 1.4 a	5.2 ± 0.2 a	192.0 ± 12.6 a
		S	32 ± 3 c	15 ± 2 c	2.1 ± 0.1 a	9.4 ± 0.6 c	0.6 ± 0.0 c	5.8 ± 0.2 b
		SM	57 ± 1 b	31 ± 1 b	2.1 ± 0.0 a	16.2 ± 0.8 b	1.1 ± 0.0 b	17.8 ± 0.3 b
	H4	CK	93 ± 2 a	68 ± 2 a	1.5 ± 0.0 c	39.8 ± 0.3 a	5.1 ± 0.2 a	202.2 ± 5.0 a
		S	38 ± 3 c	23 ± 1 c	2.2 ± 0.0 a	9.9 ± 1.2 c	0.6 ± 0.1 c	5.7 ± 1.2 c
		SM	67 ± 1 b	31 ± 1 b	1.9 ± 0.1 b	22.2 ± 0.6 b	1.2 ± 0.0 b	25.5 ± 1.6 b
NaHCO_3_ (75 mM)	H1	CK	93 ± 3 a	62 ± 2 a	1.4 ± 0.1 b	38.3 ± 0.9 a	5.4 ± 0.1 a	207.8 ± 3.2 a
		S	19 ± 2 c	9 ± 2 c	2.2 ± 0.3 a	5.3 ± 0.6 c	0.5 ± 0.1 c	2.8 ± 0.2 c
		SM	37 ± 1 b	15 ± 2 b	1.9 ± 0.2 ab	12.4 ± 0.7 b	0.7 ± 0.0 b	9.1 ± 0.7 b
	H2	CK	92 ± 2 a	62 ± 2 a	1.5 ± 0.1 c	37.4 ± 0.5 a	5.4 ± 0.3 a	201.4 ± 14.6 a
		S	21 ± 1 c	9 ± 2 c	2.2 ± 0.1 a	6.0 ± 0.1 c	0.5 ± 0.0 b	2.7 ± 0.3 b
		SM	39 ± 2 b	17 ± 2 b	2.0 ± 0.1 b	12.5 ± 0.6 b	0.7 ± 0.0 b	8.7 ± 0.3 b
	H3	CK	88 ± 2 a	61 ± 4 a	1.4 ± 0.0 b	36.8 ± 1.4 a	5.2 ± 0.2 a	192.0 ± 12.6 a
		S	18 ± 2 c	7 ± 2 b	2.5 ± 0.1 a	4.4 ± 0.5 c	0.6 ± 0.0 b	2.7 ± 0.4 b
		SM	29 ± 1 b	12 ± 2 b	2.2 ± 0.3 a	8.6 ± 1.0 b	0.7 ± 0.1 b	6.3 ± 0.7 b
	H4	CK	93 ± 2 a	68 ± 2 a	1.3 ± 0.0 b	39.8 ± 0.3 a	5.1 ± 0.2 a	202.2 ± 5.0 a
		S	37 ± 4 c	13 ± 1 b	2.0 ± 0.1 a	11.9 ± 0.9 c	0.6 ± 0.1 c	6.7 ± 0.6 c
		SM	49 ± 1 b	16 ± 2 b	1.8 ± 0.1 a	17.4 ± 0.9 b	0.9 ± 0.0 b	14.8 ± 1.0 b
Na_2_CO_3_ (30 mM)	H1	CK	93 ± 3 a	62 ± 2 a	1.4 ± 0.1 c	38.3 ± 0.9 a	5.4 ± 0.1 a	207.8 ± 3.2 a
		S	35 ± 1 c	19 ± 2 b	2.3 ± 0.1 a	9.4 ± 0.7 c	0.5 ± 0.0 c	4.4 ± 0.2 c
		SM	44 ± 3 b	18 ± 2 b	1.7 ± 0.1 b	15.6 ± 0.6 b	0.8 ± 0.1 b	12.7 ± 0.7 b
	H2	CK	92 ± 2 a	62 ± 2 a	1.5 ± 0.1 b	37.4 ± 0.5 a	5.4 ± 0.3 a	201.4 ± 14.6 a
		S	19 ± 1 c	10 ± 2 c	1.9 ± 0.1 a	6.2 ± 0.5 c	0.5 ± 0.1 b	3.2 ± 0.3 b
		SM	55 ± 2 b	27 ± 2 b	1.9 ± 0.0 a	17.1 ± 0.3 b	0.7 ± 0.1 b	12.3 ± 1.3 b
	H3	CK	88 ± 2 a	61 ± 4 a	1.4 ± 0.0 b	36.8 ± 1.4 a	5.2 ± 0.2 a	192.0 ± 12.6 a
		S	31 ± 2 c	9 ± 2 c	1.9 ± 0.3 a	10.4 ± 1.8 c	0.6 ± 0.1 b	5.6 ± 0.2 b
		SM	48 ± 2 b	27 ± 2 b	1.7 ± 0.1 ab	16.4 ± 0.8 b	0.8 ± 0.1 b	13.6 ± 1.0 b
	H4	CK	93 ± 2 a	68 ± 2 a	1.3 ± 0.0 c	39.8 ± 0.3 a	5.1 ± 0.2 a	202.2 ± 5.0 a
		S	41 ± 2 c	22 ± 2 c	2.1 ± 0.1 a	11.9 ± 1.0 c	0.6 ± 0.1 b	7.5 ± 0.3 c
		SM	69 ± 3 b	30 ± 2 b	1.7 ± 0.1 b	24.1 ± 1.2 b	0.9 ± 0.0 b	20.5 ± 0.3 b

Values are mean standard deviation (SD, n = 3). Different letters indicate significant differences by Tukey’s test (*p* < 0.05). CK, control treatment; S, salt stress without MT; SM, salt stress with 100 μM MT; GR, germination rate; GP, germination potential; MGT, mean germination days; RL, root length; VI, vigor index; GI, germination index.

## Data Availability

Data are contained within the article.

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
