# Peer review of "The Physiological Mechanism of Exogenous Melatonin on Improving Seed Germination and the Seedling Growth of Red Clover (Trifolium pretense L.) under Salt Stress"

_plants, 2024, doi:10.3390/plants13172527_

Round 1

Reviewer 1 Report

Comments and Suggestions for Authors

The manuscript concerns the effect of melatonin in mitigating salt stress in red clover. Generally, the paper is well written. However, there are some flaws that need to be corrected. The details are listed below:

L23-25: add some % changes of examined parameters between treatments

L38: correct the references in brackets. Check throughout the paper

L49: ‘soluble soil’ – rephrase

L75-77: improve the justification for this study

L179: antioxidant enzymes activity

L260-263: emphasize the ubiquitous role of antioxidant enzymes in mitigating different abiotic (pesticides, salinity, drought, etc.) and biotic (plant pathogens) stresses. For this purpose refer to https://doi.org/10.1016/j.chemosphere.2022.136284 and https://doi.org/10.3390/agronomy13051378

L345: use passive voice in Materials and Methods

L368, 390: indicate composition of the reaction mixture and solution

Comments on the Quality of English Language

Minor editing of English language required.

Author Response

L23-25: add some % changes of examined parameters between treatments

Response: Thank you very much for your comments. We have revised this part as your suggestions. They can be found in manuscripts L23-26: Foliar spraying of 50 μM and 25 μM MT solution significantly increased SOD activity (21-127%), POD activity, soluble sugar content, proline content (22-117%), chlorophyll content (2-66%), and net photosynthetic rate. It reduced the MDA content (14-55%) and intercellular CO2 concentration of red clover seedlings under salt stress.

L38: correct the references in brackets. Check throughout the paper

Response: We apologize for the mistake in the references.We have revised and corrected the error in the references in the manuscript.

L49: ‘soluble soil’ – rephrase

Response: Thanks. We have corrected the errors in this section. They can be found in manuscripts L51-53: Particularly in the north, the climatic characteristics of low rainfall and high evaporation result in a tendency for soluble salts in the soil to accumulate in the surface layer, leading to soil salinization.

L75-77: improve the justification for this study

Response: Thanks.We have improved the justification. They can be found in manuscripts L80-84: but there are limited studies on the effects of exogenous MT on red clover, and previous studies mainly focused on a single material and a single stress condition. Therefore, in this study, the responses of different varieties of red clover to different types of salt stress were compared, including not only NaCl (the most used), but also NaHCO3, Na2CO3, and Na2SO4.

L179: antioxidant enzymes activity

Response: Thanks.We have revised this part as your suggestions.This can be found in manuscripts L190.

L260-263: emphasize the ubiquitous role of antioxidant enzymes in mitigating different abiotic (pesticides, salinity, drought, etc.) and biotic (plant pathogens) stresses. For this purpose refer to https://doi.org/10.1016/j.chemosphere.2022.136284 and https://doi.org/10.3390/agronomy13051378

Response: Thank you very much for your comments. We have revised this part as your suggestions. They can be found in manuscripts L268-276: Among them, melatonin enhances antioxidant enzyme activity, which scavenges ROS, reduces lipid peroxidation, protects the integrity of cell membranes [52,53], and delays the effects of stress on plant growth and development [54]. ROS are produced by abiotic stresses (e.g., pesticides, salinity, drought) and biotic stresses (e.g., phytopathogens), which can lead to cell damage and death. Antioxidant enzymes have a universal role in mitigating different abiotic and biotic stresses and can scavenge ROS, thereby protect-ing cells from damage [55,56]. It was found that both humic acid and nitrophenol effec-tively increased CAT activity in wheat under fungal infestation and alleviated oxidative stress associated with biotic stresses [55].

[55] Iwaniuk, P.; Łuniewski, S.; Kaczyński, P.; Łozowicka, B. The influence of humic acids and nitrophenols on metabolic compounds and pesticide behavior in wheat under biotic stress. Agronomy. 2023, 13, 1378. doi: 10.3390/agronomy13051378

[56] Iwaniuk, P.; Kaczyński, P.; Pietkun, M.; Łozowicka, B. Evaluation of titanium and silicon role in mitigation of fungicides toxicity in wheat expressed at the level of biochemical and antioxidant profile. Chemospher., 2022, 308, 136284. doi: 10.1016/j.chemosphere.2022.136284

L345: use passive voice in Materials and Methods

Response: Thank you very much for your comments. We have revised this part as your suggestions. They can be found in manuscripts L363-443. (4. Materials and Methods)

L368, 390: indicate composition of the reaction mixture and solution

Response: Thank you very much for your comments. We have revised this part as your suggestions. They can be found in manuscripts L405-406: 100 μl was taken and mixed with 3.9 ml of the reaction mixture (1 mg/ml EDTA-Na2, 0.1 mg/ml riboflavin, 20 mg/ml L-methionine, 1 mg/ml NBT).

L410-411: The 40 μl of crude enzyme extract was taken and added to 3 ml of reaction solution (0.2 mol/L phosphate buffer, guaiaco, 30% H2O2

Reviewer 2 Report

Comments and Suggestions for Authors

The article addresses ability of melatonin treatment to increase resistance of red clover to salt stress in terms of germination and growth. It describes mechanisms involved in the increase in resistance to salt stress: activity of photosynthesis and antioxidant enzymes, decline in membrane injury by ROS, osmotic adjustment, maintenance of stomatal conductance and transpiration. What is really new and interesting in this work is the comparison of reactions of different varieties of red clover to different types of salt stress involving not only NaCl (which is used most often), but also NaHCO3, Ca2CO3 and Na2SO4 . Still I have some recommendations for the improvement of the article, which should be followed before I can recommend acceptance of the article. Some of my criticisms relate only to the wording, while others relate to more serious shortcomings. I arrange my comments according to the sequence in the text of the sentences that I recommend paying attention to.

1.     Lines 38-40. “It is unique in its ability to fix nitrogen in the soil, which can increase soil fertility, inhibit weed growth, and mitigate soil erosion.” – the sentences should be modified. In its present state it sounds as if ability to fix nitrogen  directly inhibits weed growth and mitigate soil erosion.

2.     Line 48. “high evaporation result in more soluble soil” – something must be wrong here. Should not it be “saline soil”?

3.     Lines 53-55. “Salt stress limits red clover growth, affecting physiological processes like cell mem-brane damage, antioxidant activity, and osmotic adjustment, ultimately reducing agricultural yield” – This sentence should be modified. Damage to cell membranes actually limits growth, but antioxidant activity and osmotic regulation, on the contrary, protect plants and support growth.

4.     Line 55-56. “Red clover responds with enzyme and non-enzyme systems, including  SOD and POD for ROS scavenging” - If examples of enzymes involved in ROS scavenging are presented, the same should be done for non-enzyme systems.

5.     Line 62. “enhancing yield and quality” – something is missing here. Quality of what is meant?

6.     Line 66.  “MT is resistant to abiotic stresses” – is chemical resistance to stresses is meant or authors wanted to tell that MT increases plant resistance to stress.

7.     Line 92. “different degrees for different species” – I think that not species, but varieties of the same species (red clover) were meant.

8.     Line 93. THIS IS THE MOST IMPORTANT REMARK!!!“H1 was the most severely inhibited” – here and throughout the whole text such statements (comparison between different varieties and salt treatments) are not supported with statistical analysis, since Duncan test was seemingly applied only for comparison between means for each variety and treatment separately. Instead, it was NECESSARY to perform one way ANOVA and Duncan test for all means in each column (each indicator). Only in this case the means for different varieties and salt treatment can be compared. Extent of inhibition or stimulation should be evaluated as percentage of decline or increase. It was calculated in some, but not in all cases. Only this approach allows statements concerning most or least severely inhibited or stimulated indicators.

9.     Line 117. “H1 was less damaged by stress” – similar remark as above.

10.  Lines 126-127. “Different letters indicate significant differences by the Duncan test” – it should be added that significant difference between means is meant.

11.  Lines 132-133. “Compared with CK, the contents of soluble sugars, proline, MDA, and relative conductivity were significantly increased in seedlings under salt stress, causing damage to the structure of the plasma membrane, which maintained osmotic balance by accumulating osmoregulatory substances.” – this should be strongly rectified. It sounds as if damage to the structure of the plasma membrane maintains osmotic balance by accumulating osmoregulatory substances. MDA and increase in relative conductivity are really indicators of damage, while soluble sugars and proline enable osmotic adjustment. These effects should not be described in one sentence.

12.  One more remark concerning this sentence. It should be explained how electric conductivity is related to membrane damage. It should be explained in M & M section how it was measured. Does it mean electrolyte leakage by  measuring electrical conductivity of the leachate from plant pieces?

13.  Line 143. “H3 showed the most significant increase in proline content” - the same remark as in point 8.

14.  Line 167. “H1 suffered the least stress damage” - the same remark as in point 8.

15.  Lines 171-173. “the damage to H1 was the least, and the SOD and POD activities of H1 were most significantly increased by 136% and 22%, respectively, after MT treatment” - the same remark as in point 8.

16.  Lines 188-189. “photosynthesis was weakened, resulting in yellowing of the plants and weakened growth” – The sentences should be rephrased. Yellowing (decline in chlorophyll) can be the cause of weakened photosynthesis and not vice versa.

17.  Line 195. “H2 was most severely damaged” – the same remark as in point 8.

18.  Line 270-271. “MT was used to cope with free radicals by increasing the activity of antioxidant enzyme systems in red clover seedlings to damage to cell membranes” – this should be rephrased. In present form it sounds as if antioxidant system damages cell membranes.

19.  Line 313-315. “The increase in intercellular CO2 concentration … under salt stress led to … growth stagnation” – authors should think over this sentence. CO2 by itself does not cause growth stagnation. It is just indicator of decreased photosynthesis that results in inhibition of growth.

Comments on the Quality of English Language

Moderate editing of English language required.

Author Response

  1. Lines 38-40. “It is unique in its ability to fix nitrogen in the soil, which can increase soil fertility, inhibit weed growth, and mitigate soil erosion.” – the sentences should be modified. In its present state it sounds as if ability to fix nitrogen  directly inhibits weed growth and mitigate soil erosion.

Response: Thank you very much for your comments. We have revised this part as your suggestions. They can be found in manuscripts L39-43: It has a strong nitrogen fixing capacity, which can reduce the reliance on chemical nitrogen fertilizers in agricultural production. At the same time, as a kind of green manure, it can be turned and buried into the soil to increase soil fertility, and its root system can penetrate deep into the soil, which can help to loosen the soil and increase the permeability, thus improving the soil structure [2,3].

[1] Thilakarathna, M.S.; Papadopoulos, Y.A.; Grimmett, M.; Fillmore, S.A.; Crouse, M.; Prithiviraj, B. Red clover varieties show nitrogen fixing advantage during the early stages of seedling development. Canadian Journal of Plant Science. 2017, 98, 517-526. doi:10.1139/cjps-2017-0071

[2] McKenna, P.; Cannon, N.; Conway, J.; Dooley, J. The use of red clover (Trifolium pratense) in soil fertility-building: A Review. Field Crops Research. 2018, 221, 38-49. 

  1. Line 48. “high evaporation result in more soluble soil” – something must be wrong here. Should not it be “saline soil”?

Response: Thanks. We have corrected the errors in this section. They can be found in manuscripts L51-53: Particularly in the north, the climatic characteristics of low rainfall and high evaporation result in a tendency for soluble salts in the soil to accumulate in the surface layer, leading to soil salinization.

  1. Lines 53-55. “Salt stress limits red clover growth, affecting physiological processes like cell mem-brane damage, antioxidant activity, and osmotic adjustment, ultimately reducing agricultural yield” – This sentence should be modified. Damage to cell membranes actually limits growth, but antioxidant activity and osmotic regulation, on the contrary, protect plants and support growth.

Response: Thank you very much for your comments. We have revised this part as your suggestions. They can be found in manuscripts L57-61: Salt stress causes damage to the structure and function of plant cell membranes, increased membrane permeability, and extravasation of intracellular substances (e.g., electrolytes, organic matter), thus affecting the normal function of cells. It also induces excessive production of reactive oxygen species (ROS) in plants, leading to membrane lipid peroxidation and further damage to cell membranes [14-17].

  1. Line 55-56. “Red clover responds with enzyme and non-enzyme systems, including  SOD and POD for ROS scavenging” - If examples of enzymes involved in ROS scavenging are presented, the same should be done for non-enzyme systems.

Response: Thank you very much for your comments. We focus on describing the enzyme system in the manuscript. We have revised this part and they can be found in manuscripts L61-62: Red clover responds with enzyme system, including SOD and POD for ROS scavenging [18,19].

5.Line 62. “enhancing yield and quality” – something is missing here. Quality of what is meant?

Response: We apologise for the misrepresentation of this section and have amended the description of this section. This section can be found in manuscripts L67: “enhance plant resistance and maintain plant growth”.

  1. Line 66.  “MT is resistant to abiotic stresses” – is chemical resistance to stresses is meant or authors wanted to tell that MT increases plant resistance to stress.

Response: We are very sorry that we did not express clearly in this part. We wanted to tell that MT increases plant resistance to stress. We have revised this part and they can be found in manuscripts L71-73: Research has demonstrated that MT possesses resistance to various abiotic stresses, including high temperatures, salinity, and heavy metals that can negatively affect plant growth and development [28,29].

7.Line 92. “different degrees for different species” – I think that not species, but varieties of the same species (red clover) were meant.

Response: Thank you for the suggestion. We have changed the word into “varieties”.

  1. Line 93. THIS IS THE MOST IMPORTANT REMARK!!!“H1 was the most severely inhibited” – here and throughout the whole text such statements (comparison between different varieties and salt treatments) are not supported with statistical analysis, since Duncan test was seemingly applied only for comparison between means for each variety and treatment separately. Instead, it was NECESSARY to perform one way ANOVA and Duncan test for all means in each column (each indicator). Only in this case the means for different varieties and salt treatment can be compared. Extent of inhibition or stimulation should be evaluated as percentage of decline or increase. It was calculated in some, but not in all cases. Only this approach allows statements concerning most or least severely inhibited or stimulated indicators.

Response: We are very sorry that our expression here is not rigorous enough.The description “most or least inhibited or stimulated” that appeared in this section has been modified, and we have provided a comparative description of the changes between treatments and controls for each variety. We have revised this part and they can be found in manuscripts L99-106: Compared with salt stress, the MT treatment significantly increased GR by 11-36%, GP by 3-17%, GI by 24-176%, RL by 17-100%, and VI by 63-284%. Among them, under NaCl stress, the GR and VI of HI significantly increased by 28% and 172%, respectively, and the GR of H4 reached 80% after MT treatment. Under Na2SO4 stress, the GR and GI of H4 significantly increased by 29% and 124%, respectively, and the RL doubled after MT treatment. Under NaHCO3 stress, GI and VI of H2 significantly increased by 108% and 222%, respectively, after MT treatment. Under Na2CO3 stress, GR, GI, and VI of H2 were significantly increased by 36%, 176%, and 284%, respectively, after MT treatment.

  1. Line 117. “H1 was less damaged by stress” – similar remark as above.

Response: Thanks.We have revised this part as your suggestions.This can be found in manuscripts L121-128: and the plant height of red clover seedlings significantly increased by 8-18%. Under NaCl and Na2SO4 stress, the plant height of H1 was significantly reduced by 39% and 31%, respectively, and significantly increased by 18% and 14% after spraying 50 μM MT, which delayed the inhibition of plant growth. Under NaHCO3 stress, the plant height of H3 was significantly improved by 15% after spraying 25 μM MT. Under Na2CO3 stress, H2, and H3 plant heights were significantly reduced by 39% and 32%, respectively, and MT treatment significantly increased H2 by 14%, and H3 by only 8% (Figure 1).

10.Lines 126-127. “Different letters indicate significant differences by the Duncan test” – it should be added that significant difference between means is meant.

Response: Thank you very much for your comments. We have revised this part as your suggestions.

  1. Lines 132-133. “Compared with CK, the contents of soluble sugars, proline, MDA, and relative conductivity were significantly increased in seedlings under salt stress, causing damage to the structure of the plasma membrane, which maintained osmotic balance by accumulating osmoregulatory substances.” – this should be strongly rectified. It sounds as if damage to the structure of the plasma membrane maintains osmotic balance by accumulating osmoregulatory substances. MDA and increase in relative conductivity are really indicators of damage, while soluble sugars and proline enable osmotic adjustment. These effects should not be described in one sentence.

Response: We are very sorry that we did not express clearly in this part. We have revised this part as your suggestions. They can be found in manuscripts L138-143: Compared with CK, soluble sugar and proline contents were significantly increased under salt stress, and the accumulation of these substances helped the plants to maintain osmotic homeostasis; the increase in MDA and relative conductivity reflected the increase in cell membrane permeability, which is usually due to the disruption of the cell membrane structure and the damage of the plant cell membrane, resulting in membrane lipid peroxidation.

12.One more remark concerning this sentence. It should be explained how electric conductivity is related to membrane damage. It should be explained in M & M section how it was measured. Does it mean electrolyte leakage by  measuring electrical conductivity of the leachate from plant pieces?

Response: Thank you very much for your comments. We have revised this part as your suggestions. They can be found in manuscripts L140-143: the increase in MDA and relative conductivity reflected the increase in cell membrane permeability, which is usually due to the disruption of the cell membrane structure and the damage of the plant cell membrane, resulting in membrane lipid peroxidation.

The method for determining relative conductivity has been added to the Materials and Methods section. They can be found in manuscripts L429-434: The relative conductivity was determined by the immersion method [96]. Fresh leaves were taken, cut into pieces, weighed 0.1 g, and put into a 10 ml clean centrifuge tube, left for 20 h, during which time it was frequently shaken up and down, and the conductivity was measured as R1 by a conductivity meter, boiled in a boiling water bath for 30 min, and then the conductivity was measured as R2 after cooling, and the relative conductivity was calculated.

[96] Prášil, I.; Zámečnı́k, J. The use of a conductivity measurement method for assessing freezing injury: I. Influence of leakage time, segment number, size and shape in a sample on evaluation of the degree of injury. Environmental and Experimental Botany. 1998, 40, 1-10.

  1. Line 143. “H3 showed the most significant increase in proline content” - the same remark as in point 8.

Response: Thanks.We have revised this part as your suggestions. They can be found in manuscripts L147-157: Compared with salt stress, melatonin treatment significantly increased the soluble sugar and proline contents of red clover seedlings by 6%-51% and 22%-117%, respectively, and the MDA contents were significantly reduced by 14%-55%. Under NaCl and Na2SO4 stress, the soluble sugar and proline contents of H1 significantly increased by 20% and 18%, and 92% and 111%, respectively, and the MDA contents of H4 were significantly reduced by 55% and 35% after spraying 50 μM MT. Under NaHCO3 and Na2CO3 stress, the proline contents of H3 were significantly increased by 117% and 114% after spraying 25 μM MT, respectively (Figures 2 and S2). The results showed that foliar spraying of melatonin could increase the accumulation of osmoregulatory substances to maintain osmotic balance in red clover seedlings under salt stress, thus delaying the inhibition of plant growth by salt stress.

  1. Line 167. “H1 suffered the least stress damage” - the same remark as in point 8.

Response: Thanks.We have revised this part as your suggestions. They can be found in manuscripts L177-180: Under NaCl stress, the SOD and POD activities of H1 were significantly increased by 64% and 38%, respectively, after the spraying of 50 μM MT, which reduced the damage of ROS to the cell membranes, maintained the normal cellular function, and reduced the damage to seedling growth.

  1. Lines 171-173. “the damage to H1 was the least, and the SOD and POD activities of H1 were most significantly increased by 136% and 22%, respectively, after MT treatment” - the same remark as in point 8.

Response: Thanks.We have revised this part as your suggestions. They can be found in manuscripts L180-184: Under Na2SO4 stress, MT treatment significantly increased the SOD and POD activities of H4 by 127% and 46%, respectively. Under NaHCO3 stress, the SOD and POD activities of H2 were significantly increased by 174% and 11%, respectively, after spraying 25 μM MT. Under Na2CO3 stress, the SOD and POD activities of H1 were significantly increased by 136% and 22%, respectively, after MT treatment.

16.Lines 188-189. “photosynthesis was weakened, resulting in yellowing of the plants and weakened growth” – The sentences should be rephrased. Yellowing (decline in chlorophyll) can be the cause of weakened photosynthesis and not vice versa.

Response: Thanks.We have revised this part as your suggestions. They can be found in manuscripts L199-201: Compared with CK, the intercellular CO2 concentration significantly increased and other photosynthetic parameters significantly decreased under salt stress, leading to a decrease in photosynthetic rate and a slowdown in plant growth.

  1. Line 195. “H2 was most severely damaged” – the same remark as in point 8.

Response: Thanks. We have revised this part and they can be found in manuscripts L206-218: Compared with salt stress, melatonin treatment significantly increased the chlorophyll contents by 2-66% and significantly reduced intercellular CO2 concentration by 15-42%. Under NaCl and Na2SO4 stress, the chlorophyll content of H2 was significantly increased by 65% and 25%, and the intercellular CO2 concentration was reduced by 27% and 24% after spraying 50 μM MT, respectively. Under NaHCO3 stress, the chlorophyll content of H1 was significantly increased by 36%, and the intercellular CO2 concentration was significantly reduced by 40% after 25 μM MT treatment. Under Na2CO3 stress, the chlorophyll content of H3 was significantly increased by 23%, and the intercellular CO2 concentration was significantly reduced by 42% after MT treatment (Figure 4). The results showed that foliar spraying of melatonin could regulate the stomatal movement and increase the chlorophyll content of red clover seedlings under salt stress, thus increasing the photosynthetic rate and reducing the inhibition of seedling growth.

  1. Line 270-271. “MT was used to cope with free radicals by increasing the activity of antioxidant enzyme systems in red clover seedlings to damage to cell membranes” – this should be rephrased. In present form it sounds as if antioxidant system damages cell membranes.

Response: We are very sorry that we did not express clearly in this part. We have revised this part and they can be found in manuscripts L287-290: MT improves the resistance of red clover seedlings by increasing the activity of antioxidant enzyme system in red clover seedlings, thereby scavenging reactive oxygen molecules, protecting cell membranes from oxidative damage, and maintaining normal plant growth and development.

19.Line 313-315. “The increase in intercellular CO2 concentration … under salt stress led to … growth stagnation” – authors should think over this sentence. CO2 by itself does not cause growth stagnation. It is just indicator of decreased photosynthesis that results in inhibition of growth.

Response: We are very sorry that we did not express clearly in this part. We have revised this part and they can be found in manuscripts L330-334: Under salt stress, stomatal closure restricts further gas exchange and increased intercellular CO2 concentration may lead to a decrease in the rate of photosynthesis, and the chlorophyll content decreases, leading to yellowing of the leaves and reducing the plant's ability to absorb light energy, thus diminishing the efficiency of photosynthesis.

Round 2

Reviewer 2 Report

Comments and Suggestions for Authors

Authors carefully addressed all my comments. I am satisfied and recommend to  accept the article in its current form

Comments on the Quality of English Language

Minor revision of English will be sufficient

Author Response

Thank you very much for your careful review and endorsement.